Alterations in fiber pathways reveal brain tumor typology: a diffusion tractography study

Campanella Martina 1 martina.campanella@iit.it
Ius Tamara 1 2
Skrap Miran 2
Fadiga Luciano 1 3
1 Department of Robotics, Brain and Cognitive Sciences, Istituto Italiano di Tecnologia , Genoa , Italy
2 Department of Neurosurgery, Az. Ospedaliero-Universitaria Santa Maria della Misericordia , Udine , Italy
3 Section of Human Physiology, University of Ferrara , Italy
Iacoboni Marco
Electronic publication date: 2014 Sep 9
Publication date: 2014
Volume: 2
Electronic Location ID: e497
Received 2013 Dec 27; Accepted 2014 Jul 6
Copyright: © 2014 Campanella et al.
Copyright year: 2014
Copyright holder: Campanella et al.
License: This is an open access article distributed under the terms of the Creative Commons Attribution License, which permits unrestricted use, distribution, and reproduction in any medium, provided the original author and source are credited.
License URL: https://creativecommons.org/licenses/by/3.0/

Keywords: Brain tumor, White matter, Fiber tractography

Funding: Fondazione Istituto Italiano di Tecnologia Funding for this study was provided by the Fondazione Istituto Italiano di Tecnologia. The funders had no role in study design, data collection and analysis, decision to publish, or preparation of the manuscript.

==============================
Conventional structural Magnetic Resonance (MR) techniques can accurately identify brain tumors but do not provide exhaustive information about the integrity of the surrounding/embedded white matter (WM). In this study, we used Diffusion-Weighted (DW) MRI tractography to explore tumor-induced alterations of WM architecture without any a priori knowledge about the fiber paths under consideration. We used deterministic multi-fiber tractography to analyze 16 cases of histologically classified brain tumors (meningioma, low-grade glioma, high-grade glioma) to evaluate the integrity of WM bundles in the tumoral region, in relation to the contralateral unaffected hemisphere. Our new tractographic approach yielded measures of WM involvement which were strongly correlated with the histopathological features of the tumor (r = 0.83, p = 0.0001). In particular, the number of affected fiber tracts were significantly (p = 0.0006) different among tumor types. Our method proposes a new application of diffusion tractography for the detection of tumor aggressiveness in those cases in which the lesion does not involve any major/known WM paths and when a priori information about the local fiber anatomy is lacking.

Introduction

A number of brain pathologies affect white matter (WM) fiber pathways, either by disruption, infiltration or displacement (Field et al., 2004; Jellison et al., 2004). Knowledge of such alterations may provide useful information during neurosurgery, particularly in the case of infiltrating lesions, where the extent of excision and the prognosis are positively correlated (Skrap et al., 2004; Clark et al., 2003). In fact, while highly aggressive lesions significantly alter morphology and impair functionality of infiltrated WM, less aggressive tumors may simply displace the surrounding brain structures. These differing behaviors influence surgical strategy, mainly aimed at finding the best compromise between amount of tissue removed and degree of preservation of brain functionality.

Nowadays, advanced techniques such as diffusion tensor imaging (DTI) allow for non-invasive tracking of WM fiber bundles characterized by a well-known hodology and course (e.g., the corticospinal system). The task becomes more difficult, however, when a priori information about the anatomy of the local fibers is lacking.

DTI is based on the fact that water diffusion is greater along fiber main axis rather than perpendicularly to it and can be used to estimate fiber orientation in each MR voxel (Basser, Mattiello & Le Bihan, 1994). Tractography uses this local information from the reconstructed diffusion tensors to identify global white matter tracts (Conturo et al., 1999; Mori et al., 1999; Basser et al., 2000). Over the last decade, diffusion MRI has often been used to investigate WM alterations. Clinicians have gained useful insights from these studies for surgical planning (Bertani et al., 2012; Zolal et al., 2012) and assessing reorganization after injury (Lazar, Alexander & Thottakara, 2006; Xu et al., 2010) or specific therapy (Kovanlikaya et al., 2011). Several reports have assessed damage to fiber tracts in relation to cerebral neoplasm types using DTI metrics, such as fractional anisotropy (FA) and mean diffusivities (MD) (Nowosielski et al., 2011; Kinoshita, Hashimoto & Gogo, 2008; Byrnes et al., 2011; Chen, Shi & Song, 2010). The majority of those works were based on a priori anatomical knowledge about fiber courses and use known anatomical landmarks to select seed regions and to reconstruct major fiber tracts (Jolapara et al., 2010; Clark et al., 2003; Yamada et al., 2003; Morita et al., 2011; Bobek-Billewicz et al., 2011).

However, WM mapping based on known normal anatomical locations as seeds might be misleading in the presence of a tumor, since the WM architecture can be deviated from its normal location and edema can mask the path of fiber tracts. A complementary approach to increase precision is to integrate fMRI information (Pantelis et al., 2010; Kleiser et al., 2010) or Intraoperative Electrical Stimulation (IES) (Bucci et al., 2013) data with tractographic data. IES is however invasive and can be performed only during surgery and not during surgical planning. Moreover, one must take into account the possible alteration of the functional responses (e.g., fMRI and IES responses) by the tumor (Giussani et al., 2010).

Since the diffusion tensor model infers the orientation of a single fiber per voxel, the DTI approach is only able to trace major tracts in the brain and cannot model complex WM architectures. Several approaches have recently been proposed to overcome this limitation and to estimate multiple fiber directions within the same voxel (Peled et al., 2006; Tournier, Calamante & Connelly, 2007; Sotiropoulos et al., 2008). One of such technique, Persistent Angular Structure (PAS)–MRI (Jansons & Alexander, 2003), starts from an economical spherical acquisition paradigm and computes a function of the sphere that reflects the angular structure of the water molecule displacement density by using the peaks of this function as fiber-orientation estimates. This particular method ensures high sensitivity to fiber crossings. Two families of tractography algorithms exist, deterministic (Bertani et al., 2012) and probabilistic (Behrens et al., 2003; Parker, Haroon & Wheeler-Kingshott, 2003). In the latter, the range of probable fiber orientations is probed with a Monte Carlo strategy and the total number of fiber trajectories passing through each voxel provides a confidence level of that voxel inclusion in the tract of interest.

Here, we propose a novel approach assessing fiber displacement/disruption caused by brain lesions that is based on PAS-MRI deterministic tractography from HARDI data. Lesions are outlined (here for patients affected by histologically characterized gliomas or meningiomas) and mirrored onto the contralateral normal hemisphere. This new location was then used as a seed region for PAS-MRI tractography of the HARDI data. The end point clusters of the tracts so obtained, as well as their back-reflection onto the affected hemisphere, are then used to generate a further set of tracts back to the seed ROI (or the original lesion ROI). By this approach we identify fiber bundles which would have presumably run through the tumor and their contralateral counterparts for comparison. All possible distinct bundles and trajectories of tumor-involved WM tracts were evaluated in relation to the myeloarchitecture, without any atlas-guided tract reconstruction.

We evaluated the capability of our technique capability to determine the typology and aggressiveness of the lesion by analyzing the relationship between the severity of tumor-induced damage and lesion-specific histological features. In particular, we compared the alterations of fiber pathways between the two types of lesions considered, meningioma and gliomas. We also tested the method by characterizing the degree of WM alteration according to their MIB-1 labeling indexes (Brat et al., 2008), a monoclonal antibody expression of the percentage of positive staining tumor cell nuclei. We used deterministic tractography via PAS-MRI as this is a straightforward, rapid technique. Alternatively, probabilistic tractography could be used, though the time demands are heavier and may not be compatible with routine clinical use. In order to assess the impact of our choice, in the present study we compared our results with those of probabilistic tractography. This latter comparison showed our approach to yield tractography results similar to those of probabilistic tractography, whilst the tumor assessments showed significant differences in tractography results between the tumor subgroups studied.

Material and Methods

Patient information

Sixteen patients diagnosed with brain tumors were considered for the study (9 males, 7 females), aged from 27 to 68 years, with a mean age of 42.62 ± 10.72 years (Table 1). On the basis of tumor histological features as determined by biopsy and according to the World Health Organization (WHO) criteria (Louis et al., 2007), the lesions studied included meningiomas in five patients (Cases M #1–5), low-grade gliomas in seven patients (Cases LGG #1–7), and high-grade gliomas in four patients (Cases HGG #1–4). The MIB-1 index (Behrens et al., 2003) was also calculated, and it ranged from 2 to 35%.

Table 1 Clinical information and outcomes in the cohort of patients.

Case no.	Age
(yrs)	Sex	Tumor location	Tumor size
(cm3)	MIB-1%	Percentage decrease in tract count	
MENINGIOMA	
M1	68	M	RH premotor motor	11.67	2	10.41%	
M2	55	M	LH parietal motor	52.99	2	21.25%	
M3	48	F	LH temporal parietal motor	36.16	4	10.00%	
M4	27	F	RH temporal motor premotor	106.25	4	32.00%	
M5	35	M	LH premotor motor	43.03	10	15.10%	
LOW GRADE GLIOMA	
LGG1	28	F	LH insular temporal	107.4	3	45.10%	
LGG2	37	F	RH frontal	13.29	4	24.68%	
LGG3	37	F	LH frontal premotor	53.38	4	28.72%	
LGG4	38	M	LH frontal premotor	98.83	4	81.25%	
LGG5	46	M	LH temporal occipital insular	43.2	9	81.81%	
LGG6	50	M	RH insular temporal parietal	46.15	9	81.85%	
LGG7	40	M	RH premotor	11.16	35	89.92%	
HIGH GRADE GLIOMA	
HGG1	41	F	RH temporal motor parietal	45.53	12	95.58%	
HGG2	51	F	RH frontal premotor motor	46.62	18	96.18%	
HGG3	32	M	LH frontal premotor	21.8	25	90.43%	
HGG4	49	M	LH premotor motor insular	43.02	30	91.95%	

This study was approved by the local institutional ethics committee on human research of the University Hospital Santa Maria della Misericordia of Udine and clinical data were acquired following the guidelines of the Department of Neurosurgery. Informed consent was obtained from all the patients who participated in this study.

Imaging protocol

Diffusion-weighted MRI data were obtained on a 3T scanner (ACHIEVA 3T, Philips, Best, the Netherlands) using a multi-shot DWI-SE sequence (FOV: 240 × 240 × 3105 mm; voxel 1.0 × 1.0 × 1.5 mm; slices acquired parallel to the line connecting the anterior commissure to the posterior commissure, no slice gap; TE = 74.8 ms; TR = 8,833 ms; NEX = 1; parallel reconstruction. Diffusion weightings were isotropically distributed over 64 directions with a b value of 1,000 s mm−2, for a total acquisition time of approximately 15 min). The MRI sessions also included T1-weighted (3D turbo-gradient-echo sequence, voxel size: 1.000 × 1.000 × 1.000 mm; 240 slices) and a Gadolinium-enhanced T1-weighted anatomical MR imaging (3D sagittal-turbo-flash sequence with TR = 2,300 ms, TE = 2,860 ms, IR = 1,100 ms, flip angle = 20°; voxel size: 1.000 × 1.000 × 1.000 mm; 240 slices, no gap between slices) was acquired, as well.

Data analysis

Data analysis was carried out using the Camino software package (Cook et al., 2006) (www.camino.org.uk), FSL (Smith et al., 2004) (http://www.fmrib.ox.ac.uk/fsl), AFNI (Cox, 1996) (http://afni.nimh.nih.gov/afni), and ITK-SNAP (Yushkevich et al., 2006), as follows.

Preprocessing

DTI scans were first re-aligned for eddy current-induced geometric distortions and head motion correction using affine registration to the first unweighted (B0) volume (Smith, 2002). Skull-stripping was then performed using BET (Smith, 2002).

Both anatomical T1-weighted scans were rigidly registered to the individual diffusion spaces using the unweighted b0-image to estimate the transformation parameters (Linear Image Registration Tool, FLIRT (Jenkinson & Smith, 2001)).

Direction reconstruction

A multi-fiber reconstruction approach was followed (Seunarine et al., 2007) and the PAS functions (Jansons & Alexander, 2003) were computed for each voxel. A reduced encoding approach was adopted (Sweet & Alexander, 2010), setting the number of basic functions in the PAS representation equal to 16. The local maxima of the PAS function corresponding to the three principal directions (PDs) were extracted for every voxel using the peak-finding algorithm implemented in the sfpeaks module of the Camino toolkit (Cook et al., 2006).

Seed

For each patient, a 3D tumoral region of interest (ROI) was manually defined by precisely tracing the contours of the tumor mass slice-by-slice (www.fmrib.ox.ac.uk/fsl/) on the MRI images. Both the unenhanced and enhanced acquired volumes were inspected so that non-tumor tissue and large vessels were avoided. Subsequently, the ROI was mirrored across the sagittal mid-plane by a procedure included in the ITK-SNAP package (Yushkevich et al., 2006) to obtain the homologous region in the healthy hemisphere. A segmentation based on a modified Gaussian mixture model as implemented in SPM (Ashburner & Friston, 2005), was performed to produce a whole brain white matter mask. We computed the intersection between the mirrored tumor mask and the white matter one using the voxel-by-voxel arithmetic calculations included in the AFNI software package (Cox, 1996). The intersection ROI was then transformed from the structural to the native diffusion space with a rigid-body registration, using the b0 image as a reference (Smith, 2002). In order to maintain the transformed masks as faithful as possible to those in the native brain images, the masks were conservatively thresholded at 0.9. Lastly, all the co-registered ROI were visually checked for precision.

Tracking—tumor out

The homologous ROI was used as the “seed” in the tractography analysis (Figs. 1A–1C). We employed a deterministic streamline tracking algorithm (Mori et al., 1999), modified in order to take into account multiple directions per voxel (Seunarine et al., 2007). Streamlines were generated from every point within the seed mask in the healthy hemisphere. The tracking algorithm starts the same number of fibers as the number of PAS peaks in each seed voxel. An anisotropy value less than 0.2, a curvature of the streamline by more than 60° across a voxel, and entrance of the streamline into an out-of-brain voxel were used as stopping criteria.

Figure 1 Method schematics.

(A) 3D tumor ROI was identified from the anatomical MRI image. (B) The ROI was mirrored across the sagittal mid-plane to obtain the homologous region in the healthy hemisphere. A segmentation was performed and the intersection between the flipped tumor ROI and the white matter mask was computed. (C) The intersection mask was then transformed from the structural to the native diffusion space and used as the “seed” in the deterministic tractography analysis. Streamlines were generated from every point within the seed mask in the healthy hemisphere and two (D1) or only one (D2) target regions were identified; (E1) In the first case the WM bundles connecting the two target ROIs were investigated. The targets were sagittally mirrored and, after seeding, the tracts of interest in the lesioned hemisphere were generated in the same manner. (E2) In the second case the homologous region mask was dilated and the streamlines connecting the dilated ROI and the target were tracked to estimate the contralateral WM tracts involved in the tumor area. Subsequently, both the target and the dilated homologous ROI were sagittally flipped and the contralateral tracts were reconstructed in the pathological hemisphere. (E1)–(E2) The tracts of interest in the two hemispheres were finally defined in a probabilistic framework following the same tracking procedures and according to the number of targets identified. Single connection probability images were thus obtained to confirm deterministic tracking results.

Target generation

Since the main interest was in the long- and medium-range connectivity alterations, an exclusion mask corresponding to a 2 cm dilation of the tumor region was created (Yushkevich et al., 2006) and used to exclude voxels directly neighboring the seed region by means of AFNI voxel-by-voxel arithmetic modules (Cox, 1996). The last ten voxels of each estimated fiber tract were taken into account. The resulting dataset of voxels was visually inspected together with the reconstructed tracks in order to estimate the number of distal clusters to which fibers ran from the tumor-based seed ROI. A k-means clustering algorithm developed in-house was applied to identify the specific centroids of these clusters. When fibers projected to two diametrically opposed brain areas, two centroids, one for each area, the ones pointed by most of the tracked pathways, were taken into account. In cases where fibers projected from the seed to nearby brain areas, only the centroid pointed by most of the projections was considered. This helped differentiate if the contralateral region acted as an interconnecting node between two distal brain area or as a seed station. Subsequently, since there are asymmetric tracts in the human brain (Thiebaut de Schotten et al., 2011), a wide ROI (10 mm-diameter sphere) around each defined centroid of interest was drawn (Yushkevich et al., 2006) in order to ensure the reconstruction of all likely fiber tracts in both hemispheres. This way, target regions were obtained having a relevant connectivity with the tumor volume mirrored in the non-affected side.

Tracts of interest

The lesion-based seeds and target ROIs, on both the affected and contralateral sides, were used in several ways to provide a complete evaluation of the impact of the tumor on WM fiber architecture.

Another tracking framework, based on a two ROI analysis (Campanella et al., 2013), was used to infer connectivity between the delineated areas of interest in both the affected and the contralateral hemispheres. We ran deterministic tractography to generate streamlines from every point within the brain mask, this time used as the “seed”. Bending angle and fractional anisotropy criteria for this second tractography analysis were the same in both procedures (60°, 0.2). After that, where diametrically opposed target areas were seen relative to the tumor (Cases M3, M4, LGG1, and LGG5), the WM bundles connecting them were investigated to estimate the fiber tracts which would have presumably run through the tumor. The two selected target ROIs were so applied as endpoint masks to prune the tracked bundles and to reconstruct streamlines connecting them in the healthy hemisphere. Finally, the target regions were sagittally mirrored and the tracts of interest in the affected hemisphere generated in the same manner as described above for the healthy hemisphere (Figs. 1D1 and 1E1).

In cases where only one target region was considered (Cases M1, M2, M5, LGG2, LGG3, LGG4, LGG6, LGG7, and HGG #1–4), to make inferences about possible distortion of the WM in the neighborhood of the tumor, the tumor-based ROIs were dilated using SPM morphology operators to 1.2 times their original size. This expanded ROI and the target area were then used as endpoints in the two ROI analysis of WM tract density. Subsequently, both the target and the dilated homologous ROI were sagittally flipped, and the correspondent tracts were mapped in the pathological hemisphere through the corresponding tracking process (Figs. 1D2 and 1E2).

Based on the results of this tractography procedure, two comparisons were undertaken: comparison between hemispheres and Comparison against probabilistic tractography, as described below.

Comparison between hemispheres

A numerical quantification of tumor-induced WM changes was assessed by counting the estimated fiber tracts of interest in each hemisphere. The percentage of decrease in tract fiber count on the pathological side was computed for every patient, taking as reference the tracts of the healthy hemisphere. Changes were assessed in relation to MIB1 index using a simple linear regression in order to investigate the relationship between severity of tumor-induced damages and lesion-specific histological features. In particular, for each case, we weighted the percentage of tracts decrease by dividing tracts value by individual patient’s estimated tumor size in order to account for the relation between amount of disruption and extension of the lesion. The tumor volume was computed from the lesion mask previously mentioned, taking into account the voxel size (Table 1).

Finally, for each patient, the reconstructed trajectories were rendered on the 3D brain and tumor ROI to better display the spatial relationship between WM tracts and lesion.

Comparison against probabilistic tractography

To verify whether the deterministic PAS-MRI process accurately determines all the WM connections and to obtain a measure about the confidence of the reconstructed trajectories, the pathways under examination in the two hemispheres were also defined in a probabilistic framework. In particular, we prepared a parametric model of uncertainty, computing a probability density function (PDF) of the diffusion orientations in each voxel for the multi-fiber population cases, according to the PAS reconstruction approach. Tractography was achieved by iterative streamline propagation through the computed PDFs for each PD estimated, using 1,000 iterations. The same tracking procedure of the deterministic framework was adopted: seeding everywhere in the brain volume and subsequently identifying streamlines connecting the ROIs. For each subject, the resulting collection of fiber tracks (i.e., across tumor–or mirrored tumor–and target ROIs) was combined to form a patient-specific connection probability image, where each voxel contains the number of streamlines that enter the voxel divided by the total number of streamlines in the patient’s collection of fiber tracts. Within patients, probabilistic and deterministic reconstructed tract volumes were visually compared in each hemisphere as well as between hemispheres, overlaying the two outputs on the same slices. For probabilistic tractography, voxels with the highest connectivity were considered those most likely to be part of the connecting bundles of interest (Figs. 1F1 and 1F2).

Results

Mapping the WM bundles of interest in the pathological hemisphere, after assessing them in the unaffected one, revealed different types of WM involvement by brain tumors which strongly correlated with the histopathological features of the lesions. Table 1 summarizes the alterations of fiber tracts, as well as some clinical and anatomical data concerning the patients. The higher the MIB-1 label was, the more fibers were found to have been destroyed, with a correlation coefficient equal to 0.83 (p = 0.0001), as shown in Fig. 2.

Figure 2 Linear regression between the pondered tumor-induced reductions in tracts and lesion-specific histological features (MIB-1 index) across all the studied cases; scatter plot shows the data and the estimated linear fitting.

Meningiomas

In patients M1 and M2, the fiber pathway depicted in the healthy hemisphere were found to be splayed into different branches in the affected hemisphere as a result of the tumor mass (Fig. 3). Nevertheless, the tractography output showed a peripheral region of intact white matter at the tumor boundary where the fiber tracts were still identifiable. Indeed, in the pathological hemisphere, the streamline algorithm did not find under-threshold anisotropy areas, and it reconstructed the connection pathway under examination. This was confirmed by comparing the counts of the estimated fiber tracts in both hemispheres. Most (over two-thirds) of the connecting streamlines identified in the healthy hemisphere were also tracked in the lesioned hemisphere (Table 1).

Figure 3 Comparative tractography study between the two hemispheres in the five cases of meningioma.

Cases M1, M2 and M5: Estimation of the fiber tracts between the main target and the dilated region homologous to the tumor in the healthy hemisphere; comparison with the contralateral lesioned architecture. In the case of M1, the ascending fibers from the thalamus to cerebral cortex and the descending fibers from the fronto-parietal cortex to subcortical nuclei and spinal cord were splayed into different branches in the lesioned hemisphere. The same alteration pattern was identified in the cortico-ponto-cerebellar tract reconstructed in patient M2, as a result of the tumor mass. The fiber tracts that leave the internal capsule ventrally and continue into the cerebral peduncles, pons and piramidal tract in the pathological hemispehere of patient M5 resulted bulky displaced, in relation to the contralateral WM architecture. Cases M3 and M4: estimation of the fiber tracts between the two target regions in the healthy hemisphere, and comparison with the contralateral lesioned architecture. The lesion mass changed the location and organization of the inferior fronto-occipital fasciculus tracked in the healthy hemisphere in both patients M3 and M4.

In the other three meningioma cases, deterministic tractography clearly suggested bulk displacement of WM tracts associated with tumor (Fig. 3). The lesion mass changed the location and organization of WM pathways but only slightly affected the coherence of fiber bundles. Indeed, in all cases of meningioma, we observed a modest decrease in the number of estimated fiber tracts in the lesioned hemisphere (10–32%), as compared to the healthy WM architecture (Table 1). The average (±standard deviation (SD)) percent difference in the number of estimated tracts between the two hemispheres across the 5 cases of meningioma was 17.75 ± 9.16%. This indicates that the underlying axonal structures have remained intact but spatially displaced.

Gliomas

Low-grade

In patient LGG1, two main connecting fiber bundles were tracked in the healthy WM architecture. The superior bundle of Cingulum, reconstructed in the homologous region, resulted as being dorsally displaced in the pathological hemisphere. Moreover, the inferior pathway was not identified at all in the lesioned area, probably as a result of tumor-induced destruction. Case LGG5 showed clear displacement of the Inferior Fronto-Occipital Fasciculus under consideration, though the tumor mass destroyed a large part of the fiber tracts. Two main connecting bundles of the Arcuate Fasciculus were also seen for patient LGG3. Here, however, the inferior pathway did not have a correspondent on the affected side. The number of fibers in the other main bundle identified in the healthy hemisphere was notably reduced in the proximity of the tumor. In the other low-grade gliomas (Cases LGG2, LGG4, LGG6, and LGG7), the tractography showed that the glioma did not change the location of the connecting pathway, but led to a reduction in the number of tracked fibers (Fig. 4).

Figure 4 Comparative tractography study between the two hemispheres in the seven cases of low-grade glioma.

Cases LGG1 and LGG5: Estimation of the fiber tracts between the two target regions in the healthy hemisphere, and comparison with the contralateral affected architecture. The superior bundle of Cingulum, reconstructed in the homologous region, resulted dorsally displaced in the pathological hemisphere in case LGG1, while its inferior pathway was not identified at all in the lesioned area. The inferior Fronto-Occipital Fasciculus, tracked in patient LGG5, was displaced as well. Cases LGG2, LGG3, LGG4, LGG6, and LGG7: estimation of the fiber tracts between the main target and the dilated region homologous to the tumor in the healthy hemisphere; comparison with the contralateral affected architecture. The Superior Longitudinal Fascicle identified in case LGG2 could not be tracked in the affected hemisphere. In patient LGG3, the inferior bundle of the Arcuate Fasciculus that connected the homologous region to the target ROI resulted destroyed in the pathological area. The Cerebellar tracts tracked in patients LGG4 and LGG6 appeared disrupted by the infiltrating tumor mass. Finally, in patient LGG7 the inferior part of Arcuate fasciculus, reconstructed in the healthy hemisphere resulted mostly destroyed in the contralateral area.

The number of estimated fiber tracts in the connection pathways studied was noticeably reduced on the pathological side as compared to contralateral WM architecture (25–90%, Table 1). Across these seven cases of low-grade gliomas, the average (±SD) percentage of tract reduction due to tumor infiltration was 61.9 ± 28.05%, indicating a high variability in their malignancy.

High-grade

Evidence of white matter tract infiltration was presented by all four patients with high-grade gliomas (HGG #1–4). The fiber bundles crossing the tumor in the affected hemisphere were extremely reduced, as compared to the contralateral hemisphere (90–96%, Table 1). Tractography results clearly demonstrated this loss of fibers with no displacement of white matter architecture, which is suggestive of tumor invasion (Fig. 5). Across the four high-grade gliomas, an average (±SD) percent decrease in tract fiber counts equal to 93.53 ± 2.78% was found, with respect to non-affected hemispheres.

Figure 5 Comparative tractography study between the two hemispheres in the four cases of high-grade gliomas.

Cases HGG1, HGG2, HGG3, and HGG4: estimation of the fiber tracts between the main target and the dilated region homologous to the tumor in the healthy hemispheres; comparison with the contralateral lesioned architecture. Tumor infiltration can be clearly seen in all the four cases. The Superior Longitudinal Fascicle in the affected hemisphere was largely disrupted in both patients HGG1 and HGG3. The Arcuate Fasciculus connecting the homologous area and the target ROI in the healthy hemisphere of case HGG2, resulted almost entirely destroyed in the tumor area. In addition, tractography in patient HGG4 demonstrated the huge loss of fiber tracts belonging to the Inferior Fronto-Occipital Fasciculus and the Cingulum reconstructed in the healthy brain contralateral area.

The difference in percentage of lesion-induced decrease in tracts between the two main groups (meningiomas and gliomas) was statistically significant (p = 0.0006).

The connection pathways as assessed by using probabilistic tractography confirmed the WM alteration patterns identified in each patient. The connectivity architecture obtained with the probabilistic approach corresponded well with that shown by our technique by deterministic tracking. In every patient the two types of connection analyses substantially coincided and no outlier from deterministic tracking were found, indicating that the deterministic choice did not compromise the quality of the results and captured all the connecting bundles. Three exemplificative cases of the comparisons between the two approaches are reported in Fig. 6.

Figure 6 Connection probability maps in the two hemispheres for three exemplificative cases.

(Meningioma, M3; Low-Grade Glioma, LGG4; High-Grade Glioma, HGG3). The probability maps, in which each voxel value corresponds to the number of streamlines that enter the voxel divided by the total number of streamlines in the input, are overlaid onto coronal and axial views of the b0 brain volume. Results showed different degrees of severity in the tumor-induced alterations of white matter tracts, as compared to the contralateral architecture. On the left and right sides, connectivity maps obtained by deterministic tractography confirm that the deterministic algorithm tracked all the important connections shown by the probabilistic analysis.

Discussion

In this study, we present a novel approach for defining alterations in WM paths induced by brain lesions by using diffusion tractography. Our method does not take into account anatomically pre-defined fiber tracts, and hence no a priori knowledge of the pathways under investigation is required.

In the proposed approach, the adoption of PAS-MRI as direction reconstruction method for multi-fiber deterministic tractography, reduced the classic limitation of the diffusion tensor model and its inability to differentiate tracts in cases of WM fiber crossing, branching or fanning. Our choice overcame the underestimation of the extent of tracks of interest that may lead to unreliable and clinically misleading information (Farquharson et al., 2013), although in a clinically compatible acquisition time.

The results for meningioma show displacements of the WM pathways, indicating the presence of intact but spatially deviated axonal structures. Meningioma patients maintained a degree of connectivity sufficiently comparable to the contralateral hemisphere. Indeed, brain tumors like meningiomas do not usually infiltrate the surrounding brain tissue, but simply compress or distort it (Wei, Gang & Mikulis, 2007; Carvi et al., 2010). In patients with low-grade gliomas, a mixed pattern of tract deviation and disruption was seen, again, consistent with previous studies (Kuhnt et al., 2013; Castellano et al., 2012). All the analyzed cases share a common feature. Low-grade gliomas are usually characterized by an extensive, diffuse infiltration of tumoral cells that preferentially invade the brain along myelinated fibers in white matter tracts (Duffau et al., 2004). The mass effect of the lesion bulk appeared to be insufficient to account for this reduction in fiber tracts, which should most likely be considered as an index of WM disruption caused by edema/tumor infiltration. In the presence of high-grade gliomas, tractography analysis showed an almost complete disruption of the fiber bundles (Gulati et al., 2009; Chen et al., 2012).

Our method used the percentage of decrease in tracts as a measure of the severity of WM alterations caused by the tumor mass. This data correlated with histopathological tumoral features. Indeed, our data presented significantly different degrees of WM involvement in the two groups of patients (glioma and meningioma). In addition, a strong correlation was found between the amount of fibers disrupted and the MIB-1 index.

In the proposed technique, we adopted deterministic tractography for both speed and simplicity of use. The technique could, however, be adapted to use probabilistic tracking, which is an optimal method for modeling uncertainty, generating multiple curves from a seed point. This method ensures a greater robustness to the image noise (Behrens et al., 2003).

As part of our evaluation, we compared the results of our technique with those obtained using a probabilistic framework. The latter step of the analysis was so time-consuming (requiring several days for each case) that it could not be applied in a preoperative routine setting, though this may change with hardware and algorithmic improvements. In our specific cohort, the results of probabilistic tractography showed our deterministic results obtained with PAS-MRI to yield substantially equivalent results. The consistency of our results with those of probabilistic tractography supports the assessment of differences in various types of tumors by observing WM pathway changes and suggests the feasibility of the proposed method in those cases in which the lesion does not involve major/known WM pathways and a priori information about the local fibers anatomy is lacking.

Limitations of the Study

We are aware that our work is based on a relatively small sample size. Nonetheless, the results obtained in this preliminary study and their congruence in the various patients belonging to the three main classes of tumors, suggest stereotypical tumor-induced WM alterations, in relation to the type and the severity of the lesion.

The inhomogeneity of tumor locations and sizes creates a difficulty in that some tracts may be intersected by the tumor in mid-tract, whilst others effectively extend in only one direction from the tumor area. To accommodate these two contexts, we incorporated a dual ROI tracking approach between target ROIs when apparent tract passage was seen in the original tracking from the homologous tumor ROI, and used the target ROI and the tumor (or tumor homolog) ROI for tracking otherwise. The choice as to which approach to apply, however, was made on a subjective basis and a set of objective criteria for determining which one would be desirable to perform in a given case. It is unlikely that this aspect of our technique influenced the comparisons performed as the methods were applied bilaterally for comparison at the patient level, and were mixed across the patient groups.

In the present work, we didn’t take into account pre- or peri-operative functional localization findings. When such information is available, either from fMRI or from intra-operative cortical/subcortical electrodes, the computation speed of the deterministic approach (or the recalculation of tract probability distributions) could facilitate the incorporation of such data into surgical guidance systems (Chen et al., 2012).

We used five software packages to analyze our data in order to take advantage of the specialized algorithms specific of the various packages. In view of potential use in clinical routine, a uniform interface that combines all the processing steps within a single framework would be useful.

Multi-shot echo-planar imaging (EPI) was adopted. This achieve higher spatial resolution to fully capture the complex axonal configurations and limit the partial volume effects enhanced from the anisotropic-voxel acquisition performed, at the expense of more susceptiblity to motion.

HARD tractography methods, such as the one we used, greatly improved resolution for detecting crossing fibers, but are still susceptible to errors introduced by uncertainty at sites of fiber crossing. This is typical of any deterministic tractography analysis especially with T2-weighted signal abnormalities in the proximity or within the lesion. Comparison with a complementary probabilistic analysis showed that this was not a significant problem in our cohort of patients.

Conclusion

Diffusion imaging has become an essential part of MRI examinations in brain tumor patients. To date, diffusion tractography is the only imaging technique able to provide spatial maps of WM pathways in vivo. Our method suggests that analysis of connectivity can be used to complement fractional anisotropy in clinical studies as it might reveal other features, like the deviation or disruption of fiber tracts, and correlates well with the lesion histopathology. The proposed method is able to exploit streamline tractography and may represent a possible preoperative diagnostic technique. Moreover, the inspection of the results can help in identifying WM tracts particularly important for brain functionality. Involvement of white matter fibers represents an important piece of information to correctly plan the surgical approach and to evaluate the extent of a safe resection in patients with intrinsic brain tumors. Further research is however needed to fully assess the clinical relevance of this approach.

Additional Information and Declarations

Competing Interests

Author Contributions

Human Ethics

Luciano Fadiga is an Academic Editor for PeerJ.

Martina Campanella analyzed the data, wrote the paper, prepared figures and/or tables, reviewed drafts of the paper.

Tamara Ius and Miran Skrap conceived and designed the experiments, performed the experiments, contributed reagents/materials/analysis tools, reviewed drafts of the paper.

Luciano Fadiga conceived and designed the experiments, wrote the paper, reviewed drafts of the paper.

The following information was supplied relating to ethical approvals (i.e., approving body and any reference numbers):

Az. Ospedaliero-Universitaria Santa Maria della Misericordia of Udine approved the research:

Protocol Number 61144.

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
