# Peer review of "Alterations in fiber pathways reveal brain tumor typology: a diffusion tractography study"

_PeerJ, doi:10.7717/peerj.497_

## Round 0.1 · original submission · Major Revisions

The reviewers made a number of substantial comments. I suggest you to carefully revise the manuscript by thoroughly addressing these concerns. I am confident that such careful revision process should lead to a major improvement of the manuscript.

·

Basic reporting

The overall structure of the paper is fine and the general preparation is adequate. Unfortunately, the authors have not done a satisfactory job of communicating their message.

The introduction provides an acceptable general introduction, but fails to clearly identify why an alternative is needed to existing approaches. A couple of points regarding possible displacement of tracts by lesions, interference of lesions with functional responses, and the possible lack of involvement of one or more major WM tracts would be useful.

The last paragraph of the introduction would benefit greatly from separating the outline of the method from the brief description of how the method is to be evaluated. Moreover, the rationale of the method needs to be outlined in a suitable manner because the methods section breaks the process into steps that are difficult to follow the logic of the process. The rationale for and importance of the probabilistic evaluation should also be better stated.

Experimental design

The method is reasonably described, though the sub-section on
tracts of interest was not sufficiently clear for me to understand how the number of centroids generated was decided.

While informed consent is stated, there is no indication that the study itself had been approved by the responsible ethical committee or research review board.

Validity of the findings

The results section needs fairly extensive rewriting. In particular, there are numerous comments that speculate as to the reason behind individual observations (e.g. "probably as a result of the tumor mass", and several other sentences involving "probably"). All of these comments should be eliminated, and such speculation restricted to the discussion (e.g. This could indicate that the underlying axonal structures have remained intact but spatially displaced) with relevant literature cited to support the claims or suggestions. When talking about the various pathways found, it might be useful to indicate where the lesion was and where the observed tract end point centroid(s) were.

The discussion also needs considerable revision, firstly to provide a more readily followed line of thought, and more to reflect on its place in context of existing literature. As part of the connection to the literature, it would be good to reflect evidence from elsewhere that disruption is particularly consistent with tumor cell infiltration and not simply edematous changes. There is a large literature on individual diffusion parameters in brain tumors, some relation with how the changes reported in those studies relate to tractography could be attempted.

As far as the validity of the results, the small number of cases in different conditions is a limitation, but also provides an initial indication that to some extent the type and severity of a tumor may have stereotypical impacts on WM.

Additional comments

There are a large number of linguistic peculiarities and greater care is needed in providing a consistent and complete message.

Title
because DWI is commonly used for non tensorial imaging, replace DWI with Diffusion


Abstract

In several points, the abstract is vague or imprecise where a bit more clarity would serve.

what is meant by degrees of severity in WM involvement? - there are numerous possibilities

Decreases in tumor-induced to Tumor-induced decreases
but what exactly is decreased? number of fibers, density of fibers, FA...?
a reader of the abstract may not know what has been done in the article.


Introduction

quoad vitam - probably can be deleted

Nowadays, a to A

Nevertheless, the situation becomes much harder
to
The task becomes more difficult however,

Among these, starting from ... computes
to
One such technique, Persistent Angular Structure (PAS)-MRI, starts from an economical spherical acquisition paradigm and computes

First sentence of next paragraph not necessary.

Several research ... to fiber tracts
to
Several reports have assessed damage to fiber tracts

Nevertheless, the ... DTI
to
The majority of those works dealing with tractography

massive fiber to major fiber

Alternative perspectives integrated
to
An alternative is to integrate

Materials and Methods

). They ranged in age from 27 to 68 years of age,
to
), aged from 27 to 68 years,

Classification of the lesions was
to
Lesions were classified


ref for WHO criteria

All the clinical to All clinical

by a 3T scanner (ACHIEVA to on a 3T scanner (Achieva

isotropically distributed diffusion weightings were used along
to
Diffusion weightings were isotropically distributed over

approximately equal to
to
of approximately

included patients' to included

MR. Gadolinium-enhanced to , and a contrast enhanced

according to the following procedure to as follows

DWI to DTI

which SW was used for realignment of DTI data

as implemented in BET to using BET

All anatomical T1-weighted data
to
Both anatomical T1-weighted scans
(was this necessary, or would the post-contrast have served?).

from the anatomical contrast-enhanced
to
on the contrast-enhanced anatomical

flipped across the sagittal plane
to
mirrored across the sagittal mid-plane

warped ... rigid-body - this seems a contradiction as by definition a rigid-body transformation doesn't involve warping.
was this another transformation or the ones mentioned above


the "seed" in the tractography analysis. this covers a bit much and according to my understanding of the procedure the seed is relevant only to the Tracking and Target Generation steps, and visualization, not for the actual analysis of the tracts.

which variation of anisotropy was used as a stopping condition

across the voxel to across a voxel

The "Targets" paragraph (suggest Target Generation, or Secondary Seed Generation) needs some work to clarify the process. From the manuscript to this point, I did not understand why the main interest is in the farthest projections. It seems that something is needed to express what this interest is.

used to discard voxels to used to exclude voxels

visually inspected for accuracy
to
visually inspected for their correspondence with the identified tracks

was two and a .... Figure 1.

was two. A k-means clustering algorithm developed in-house as applied to identify the specific centroids towards groups of fibers from the seed projected. A 10 mm diameter sphere around each centroid of interest was taken to define the corresponding target region (Figure 1).


I found the tracts of interest part confusing, in effect, one form of analysis applied when a long-range tract was involved, and a second for localized tracts. I would expect the preceding steps to produce a variable number of centroids (the subjectivity of the decision about number of centrois should be mentioned in discussion), but why exclude smaller centroids, how large is a brain area?
As reflected in Figure 1, why is the relatively short track set whose extensions are indicated by the central two arrows in
All of this about the limiting of the numbers of target regions should be described in the target region sections. The tracts of interest section could then simply state that where multiple targets were present these were treated as endpoints for tracking, while where only a single target was defined, the original tumor and homologous ROI were dilated and used as the second endpoint (implications of this difference in tracking strategy to be mentioned in discussion).

After all, the... to Finally, the ....

The target regions were mirrored sagittally and the tracts of interest in the lesioned hemisphere generated in the same manner as described above for the healthy hemisphere.

The means of comparing the deterministic and probabilistic tracking results is not clear (as a discussion point, what about using the probabilistic tract generation and then using the tumor and homolog seeds or their dilated versions) as include masks to evaluate the effect (hasn't somebody done this already?)).


Results

from the first paragraph, only the last sentence is needed.

resulted as being fragmented to were found to be splayed

Indeed in the pathological ... areas - by the definition of the tracking algorithm this is true, no?

Do tracts not progress on the tumor side in some cases because the anisotrpy there is too low, or because they get squashed out or diverted into non-interesting areas that

Move the sentence about visualization to the Materials and Methods section and describe the basic details of the process.

note that in the figure for M3, at least one of the tumor side tracts is wildly divergent from those seen for the homolog, and almost certainly artefactual. Given that much of the interpretation of the rest of the study relies on tract reconstruction being realistic, such a result merits comment.

counting and comparing the to comparing the counts of

Most of the connecting add (over two-thirds)

pointed out to suggested

induced by the to of WM tracts associated with

without affecting the coherence of fiber bundles - this is difficult to accept in light of the very loose structure of the fiber bundles in M3 and M4.

what does sufficiently comparable mean?

average percentage difference in what is being referred to in the last sentence on the meningioma section?

LGG3, the lesion ... the inferior pathway that ... target ROI. These fiber tracts did not ... lesioned area.
to
LGG3, the inferior pathway ... target ROI did not have a ... lesioned hemisphere.

In addition, the other
to
The number of fibers in the other

but probably destroyed several tracts, which therefore could not be

but did lead to a reduction in the number of tracts that could be

The first few sentences of the last paragraph on LGG to discussion.

The lesion mass ... healthy hemisphere. - delete

lesion mass infiltration caused - delete

This step ... routine setting. - to discussion

induced damages to related changes


much of the paragraph: In order to investigate ... should be moved to the methods section in order to complete the description of how the results of the process of tracking were evaluated in respect to tumor size and Mib1. The results from this paragraph could then be moved to the start of the results section to provide a general overview before discussing the different tracking findings in detail.

performed. In particular, for .... the percentage of decrease in tracts ... tumor size
to
performed between the percentage reduction in tract count and tumor size,

how was the probabilistic - deterministic result compared? "mostly corresponded" does not give much information - the tract colors were almost identical, the locations were similar, no other tracts found involving the tumor or homolog?

s


Discussion

suggest rewriting first paragraph.
In this study, we have presented an approach to defining alterations in WM paths induced by brain lesions that can be used in clinical cases through the use of diffusion tractography of brain fibers. The method does not examine pre-defined fiber tracts, and hence no a priori knowledge of the pathways under investigation was required, thus avoiding inter-subject variability bias. Moreover, in adopting PAS-MRI as the direction reconstruction method for multi-fiber deterministic tractography, reduced the classic limitation of the diffusion tensor model; its inability to differentiate tract in cases of WM fiber crossing, branching or fanning.

Our results in the cases of meningioma illustrated displacements of the WM pathways, indicating the ... In patients with low-grade glioma, a mixed pattern of tract deviation and disruption was seen. In the presence of high-grade glioma however, tractography was characterized by an almost complete disruption of the fiber bundles.

separate the comments on probabilistic tracking into a new paragraph, incorporating the time issue, and so on.

We are aware that our to Our

its heterogeneity. to the heterogeneity of patients.

However, to Nonetheless, this

Conclusion

3D rendering to spatial maps


Table 1

H not needed for Hand

Percentage of tracts decrease to Percentage Decrease in Tract Count

Figure 1 - strongly suggest a series of images that follow the progression of the whole process:
a) tumor ROI on 3DT1
b) transform to DTI space
c) mirror to form homolog
d) tracking and endpoint centroid definition
e) mirroring to have both homolog and tumor endpoint centroids
f) second tracking process
g-h) probabilistic tracking for comparison....

Figure 2

do the coronal views indicate both the seed and the exclusion zone around it?

Figure 5

could both the deterministic and pathological brains be displayed at the same size.


Figure 6

pondered? not


Ref

1 Pat-terns to Patterns

the use of et al. does not appear to be consistent.

39 volume and issue details

Reviewer 2 ·

Basic reporting

No Comments

Experimental design

No Comments

Validity of the findings

No Comments

Additional comments

This is an interesting study that attempts to explore the effects of different types of brain tumours on brain connections using diffusion MRI and tractography techniques. The paper is nicely written and reads well. I only have some comments that I think would clarify some sections.

P. 2: The authors use HARDI and PASMRI. Even if there is nothing wrong with that, it may sound as an overkill for patient data. The choice becomes clear later in the manuscript and in the discussion, but it could be clarified and justified here as well. Maybe the authors could briefly mention that despite other techniques being available for resolving crossings using low angular resolution data (e.g. (Peled et al, MRI 2006), (Tournier et al, NeuroImage 2007), (Sotiropoulos et al, JMRI 2008), etc) the particular choice ensures higher sensitivity to fibre crossings.

P. 2: Probabilistic Tractography: Please provide references (e.g. Behrens et al MRM 2003, Parker et al JMRI 2003).

P. 3: The choice of the imaging protocol is suboptimal. Voxels are anisotropic (suboptimal for tractography, as signal will be lower and therefore uncertainty will be higher along the smaller voxel dimensions). Also, multi-shot acquisitions are more susceptible to subject motion, which is clearly an issue for patients. The authors need to justify these choices and/or at least discuss them as limitations. Finally, the reported resolution is probably interpolated. What is the native resolution of the data? (e.g. 120mm/70 slices does not give 1.5mm that the authors report as slice thickness)

P. 4: Using five packages to do deterministic tractography is clearly unnecessary. Nothing wrong about that, but you make your life more difficult. Most of the functions you have performed using AFNI/SPM for instance can be done using FSL or SPM alone.

P. 4. References 9 and 35 are repeated, they are the same.

P. 4. Enhanced / Unenhanced: You mean Gadolinium-enhanced T1? Please clarify.

P. 5-6: Defining tracts of interest. I feel this is an awkward way of defining tracts. Tractography can estimate paths everywhere, so what do the estimated paths in terms of anatomy and their relationship to the tumour area (particularly the ones identified between two targets)? Could you give us some insight behind this choice as opposed to focusing e.g. on specific tracts defined via strictly anatomical criteria?

Figure 5: For the Meningioma case the shown example does not clearly support the main conclusion (i.e. that tracts are displaced by the tumour). Could you please provide a probabilistic tractography result for cases M3 or M4, rather than M2? That would make your point stronger and clearer.

Reviewer 3 ·

Basic reporting

The submission adheres to PeerJ policies and it is written in good English requiring only minor editing by a native English speaker.
The introduction and background sufficiently demonstrate how this work fits into the field of using MR diffusion imaging for brain surgery.

The description of the methodology is complex and it should be less redundant. The authors should try to reduce it in length.

The tables and figure adequately illustrate the findings as described in the results section.

Experimental design

The clinical research question is broad and not very well identified.
The authors aim to define alterations in WM paths induced by neoplasms.
They conclude that decreases in tumor-induced tracts were significantly different between lesion types. However they do not consider a reference index (intraoperative electrical subcortical stimulation for instance). They do not compare their findings with other imaging parameters such as volumes of abnormally appearing white matter.

The methodological approach chosen by the author for determining white matter damage induced by intraaxial and extraaxial brain neoplasm is original and sufficiently interesting.
The authors made a big effort to use a novel approach with "user-independent" semi-automatic definition of seed and target ROIs, however their only minimal supervised method is going to leave out (or miss) several white matter tracts within and around the mass that may have an important role in brain function.

The authors stop short from identifying the name of the major bundles that are infiltrated/damaged by the lesion.
The authors do not indicate the name of the major tracts identified by deterministic tractography. This step is important and it would have important implications for presurgical mapping.

It is well known that meningiomas displace and do not usually infiltrate or invade brain tissue. It was not a surprise to find that the average percentage of tract reduction due to tumor infiltration/invasion was much lower in meningiomas than in LGG and HGG.

The finding that "the higher the Mib-1 label was, the more fibers were missing" is likely a corollary finding rather than an important one. This information could be easily gather from inspection of T2WI and T1WI post-gadolinium images.

Validity of the findings

The choice to use the PAS functions for multi-fiber reconstruction was a good one, because this method is less sensitive than classic DTI to missing white matter tracts in areas of crossing, branching or fanning. However this method has its own limitations that the authors should acknowledge in the appropriate section of the discussion.

The method used to pre-process the diffusion imaging data, to identify seed and target ROIs is well described and relatively sound.

The selection of the targets appear to be partially bias. In areas of T2WI signal abnormality diffusion imaging may miss existing fibers. There are asymmetric tracts in the human brain (i.e. arcuate fasciculus). How do the authors controlled for these two issues? They should address this issue in the M&M section.

The results support the conclusions made by the authors. This reviewer ask if we really need such a complex and long methodology to quantify already well known differences in percentage of white matter damage induced by meningiomas and gliomas.

A major limitation of this study was the lack of a reference index such as intraoperative electrostimulation subcortical mapping.

Additional comments

Specific comments

Materials and methods
The authors should make an effort to identify the name of the main tracts reconstructed with MR tractography. They should report tract names in the table and figure captions.

Results
P12.8 LGG are known to infiltrate white matter tracts while HGG tend to dislocate them and eventually destroy them. In the discussion the authors should make a comment about what they found and what it is reported in the literature.

The Discussion is weak and very brief. The authors should put their finding in contest of the literature. They should describe in greater detail potential clinical implications of their findings on planning surgery in patients harboring neoplasms infiltrating eloquent areas.

In the Limitation paragraph the authors should acknowledge limitation of tracking with PAS in areas of T2-weighted signal abnormalities. They should also acknowledge possible bias of their methods in ROI target selection. Main WM tracts coursing in the proximity or within the tumor may be missed with the approach used in this study.

Figures
Please indicate name of main white matter tract reconstructed according to anatomic position and describe relationship with the tumor.

---

## Round 0.2 · Minor Revisions

While the revised manuscript is much improved, I believe the reviewer's suggestions for a minor revision should be followed to further improve the quality of this paper.

·

Basic reporting

Although the manuscript is somewhat improved, on re-reading it for this review, I cannot help but see there are many points on which the details are not sufficiently clear, nor the context adequately set. Please see the general comments below.
The figure outlining the technique does not succeed in conveying its message, an improved caption and a more linear structure would likely help. The other figure captions would also benefit a bit more descriptive nature.
The introduction has some gaps in placing the work in respect to prior literature - specifically in regards to the place of probabilistic tractography.

Experimental design

The work is original, and provides a practical alternative to existing pre-surgical analyses of DTI data based on simple visual perception or pre-existent atlases of fibre tracts.
Unfortunately, there is still room to improve the expression of the technique they propose, and separate this from the assessment procedure covered by the study.
A point that would be of interest and is not addressed is the reproducibility of the results - does it matter if the original tumor ROI is too big / a bit small? The choices - a 2cm exclusion zone, and factor of 1.2 for the tumor zone region; do they have a significant impact on the results?
The selection of what the authors call target clusters is rather ambiguous - more concrete criteria would be desirable.

Validity of the findings

As a demonstration of the methodology, the work has been adequately executed, and I have no major concerns on this front. However, the number and mix of patients do not speak to a statistical power driven study for the comparison of patient sub-types performed. as such, the results can at best be expressed as suggestive. Moreover, the discussion is very internally focussed and could be somewhat reduced making space for a more universal consideration of the technique
- are the results any different than what would have been / was seen using conventionl DTI analysis techiques?
- given the exclusion zone, and the mass effect on the tumor side, is it fair to use the argument in the introduction that a technique is needed to deal with non-major tracts, when the results reached seem likely to be dominated by major tracts - were the tracts identified by the technique really ones that could not be obtained using a tract atlas?
- are inherent laterality differences adequately addressed?
For future work, ore interesting comparisons can be envisaged, regarding different grades, types of deficits in the patients, and histological and molecular traits of the tumors.

Additional comments

Abstract

how do you know that anything in tractography is able to "properly evaluate the type and severity of WM involvement"?

suggest

… approach yielded measures of WM involvement which were strongly … In particular, the number of fiber tracts


those critical cases to those cases


Introduction

line 19 and elsewhere, please be consistent about the inclusion or not of a space preceding reference superscripts.

l22 significantly impair … functionality to significantly alter the morphology and impair functionality

l24 removal and preserved to tissue removed and degree of preservation of

l26 Nowadays to A

l 27 of fiber to of WM fiber

l 28 however more difficult to more difficult however,

l30 move ref 5 o later in sentence

l41 misleading, to misleading in the presence of a tumor,

l45 it should be taken to one must take

l45-6 the functional responses, both for fMRI and IES, induced by the tumor
to both functional MRI and IES responses by the tumor.
what is IES?

l47 one single to a single

l50 of such techniques to such technique

l52 molecules to molecule

l53 estimate to estimates

l54 higher - implies a comparison - to what are you comparing?
some information on the probabilistic approaches should be included here.

l55 assess to assessing

l56 by using to that is based on PAS-MRI

I56 - personally I consider HARDI a more widely used and complete abbreviation for the acquisition method.

l57 (HARD) data … patients affected to (HARDI) data. Lesions are delineated (here for patients affected …

l58 meningiomas. PAS … reflecting on to meningiomas) and mirrored onto

l59 the site … this new location to This new location was then

l59 seed region, to for PAS-MRI tractography of the HARDI data. The end point clusters of the tracts so obtained, as well as their reflections onto the lesioned side of the brain are then used to generate a further set of tracts back to the seed ROI (or the original lesion ROI).

l59-61 all possible … reconstruction. - move this to be the final sentence of paragraph.

l61we have been able to to we

l62 and the estimated fiber … hemisphere to and their contralateral counterparts for comparison.

l67 damages to damage

l67 studies the differences in the to compared

l69 We tested to We also tested

l71 tractography as a straightforward to tractography via PAS-MRI as this is a straightforward

I72 but we also performed … incorporating to Alternatively, probabilistic tractography could be used, though the time demands are heavier and may not be compatible with clinical use. In order to assess the impact of our choice, As part of this study we compared our results with those of probabilistic tractography. This latter comparison showed our approach to yield tractography results similar to those of probabilistic tractography, whilst the tumor assessments showed significant differences in tractography results between the tumor subgroups studied.

- the authors used "important connections" - how was / would this be assessed?
- a description of probabilistictractography earlier in the introduction could aid in highlighting potential advantages and weaknesses of the proposed technique.

Materials and Methods

I still find the description lacking a bit in "flow", and a clinical colleague I asked to read this section produced a description which suggested to me that he had not grasped the crux of the method. I would encourage the authors to make another attempt at writing this section for dummies like me.

what is the MIB-1 index, what relevance does it have, could there be a reference for it?

reverse order of sentences "Lesions included …. " and "Lesions were classified…" e.g.
On the basis of tumor histological …. criteria, the lesions studied included ….

l87 in this clinical investigation to in this study.

l89 Philips to Philips, Best, The

l91 70 slices to with slices

place all the diffusion MR sequence details in one parenthetical statement.

It appears the pre and post-contrast T1 wtd scans were identical other then the presence of contrast agent. Probably needs the scan details only once with suitable wording of the sentence(s).

Does the journal style use signed superscripts or the "/" symbol for indicating denominator units of measure?

l104 for Eddy to for eddy

l105 unweighted volume to unweighted (B0) volume

l108 individual diffusion spaces to individual's diffusion space

Regarding the use of terms seed and target, because the targets are subsequently used in effect as the seeds of interest, I wonder if a suitable phrasing (e.g. "tumor-based seeds" and "distal seeds") might make it simpler to follow the progression.
Related to this, while Figure 1 is a welcome addition, it would benefit from a more consistent labeling of the steps and uniquely associating colours of ROIs with the various seed ROIs. One should also mention that these are simplified procedures as ROI dilations, exclusion zones etc are not mentioned. Please rework the figure until someone not associated with the study is able to look at it and have a pretty good idea of the technique. A separate or extended figure might then explain the study components (tumor type evaluation, laterality evaluation, verification against probabilistic model etc…

l118 on the MRI images to on the 3D T1-weighted MR images

l119 Before a ROI was drawn to Both

l119 were both to were

A point for discussion is that with heavy deformation of the brain, even distant tract endpoints might be altered. Related to this, if one can only really look at distant connection sites, are there so many long-range fiber tracts that aren't present in atlases - arising from one of the points in the introduction about major tracts not being the full story.

l122 segmentation framework based to segmentation based

l123 and the … identified to to produce a whole brain white matter mask.

l128 the volume of … region to transformed masks

l131 "seed" to "tumor-based seed" as per comment above.

The sentence The homologous ROI … could be moved to be the first paragraph of the following paragraph.

l133 Tracking to Tracking - tumor out

l139 Target Generation to Distal Seed Generation

The sentence "The last ten…" could be moved to after the comments about exclusion zone.

l142 concerned the … investigate the to was in the

l143 exclusive mask to exclusion mask

l143 correspondent to … around) to corresponding to a 2cm dilation of the tumor region

l146 for their correspondence with to together with

l147 of clusters to distal clusters to which fibers ran from the tumor-based seed ROI.

l149 specific centroids .. projected to centroids of these clusters.

There is mention of fibers arriving at the centroids - presumable they arrive at the clusters associated with the centroid.

I didn't understand the need to differentiate between dual opposed and single clusters, perhaps a bit of explanation why this was necessary / useful. Perhaps a few words of explanation. Were there tumor ROIs that led to more endpoint clusters? If the reason for the information was to support the "tracking through the tumor region" issue, please consolidate the pieces relating to this issue in one place AFTER finishing to describe the proposed method. It is complicated enough as it is, without adding sub-tests into the mix.
A bit of justification seems missing.

l149-50 reached by most of the tracked pathways - how was this assessed?

In this paragraph, regarding the centroids, as a centroid describes a point here, are you really talking about tracks reaching the centroid, or about tracks reaching the associated cluster?

l152 a rather wide to . A wide

was 10mm bigger than all the clusters encountered? could the size be adjustable based on an estimate of the cluster volume?

l154-5 The corresponding target regions were so obtained. to In this way, distal seed regions were obtained having a relevant connectivity to the mirrored tumor volume in the non-tumor side.

paragraph starting l157-8 This paragraph in particular marks where I feel the manuscript creates confusion. I believe the details about two target regions could be moved to results. Instead, please provide a suitable introduction which relates to the proposed method and not its evaluation, or make it clear why you are distinguishing (to me pointlessly as it seems they all should have their place in a consistent evaluation of ALL the tumors) between single/ dual,near / far tracking without having provided justification for such in the introduction. As it just seems a mash of steps.

l160 another tracking framework to tracking

is this other tracking actually any different than the technique (i.e. the framework) used previously? If so MUCH more information is needed about the difference. If it is only that tracking is now done based on the

Once the distal seeds are identified, might one summarize the further steps as:

The tumor-based and distal seeds on both the affected and contralateral sides were used in in several ways to provide a complete evaluation of the impact of the tumor on WM fiber architecture. Where diametrically opposed distal seeds were seen relative to the tumor, the two distal seeds were used to infer connectivity between the areas in both the affected and contralateral hemispheres following a two ROI approach. To make inferences about possible distortion of the WM in the neighborhood of the tumor, the tour based ROIs were dilated using SPM morphology operators to 1.2 times their original size. This expanded ROI and all distal ROIs were then used in a two ROI analysis of WM tract density. Bending angle and fractional anisotropy criteria for the two ROI analyses were the same in both procedures (values). Based on the results of this proposed tractography procedure, three (or more?) comparisons were undertaken: Comparison between hemispheres, Comparison between tumor types, and Comparison against probabilistic tractography as described below.

- done in this way, the details of how many had two distal ROIs, and how many not, can be shifted to the results.

l173-8 the authors use the term contralateral first to indicate the healthy side (contralateral to the tumor) and within the same paragraph to refer to the tumor-bearing side (contralateral to the "targets"). A clear choice of terms for the two sides would avoid the need for these additional mental gymnastics on the part of the reader.

l180 WM damage - are you sure it is all damage - suggest WM change

l181 in tracts was to in tract fiber count on the pathological side was

l183 Moreover, for each … was to These changes were then assessed in relation to MIB1 index using a simple
linear regression and tumor type with WHAT STATS DONE FOR GROUPS?
l186 move the sentence "For each case … extension of the lesion" to end of l182 after healthy hemisphere. SIMPLIFY, and CLARIFY the sentence if (and it is ) possible.

l188-9 delete sentence "Next we…"

l189 patient, reconstructed to patient, the reconstructed

l190 shown in 3D render brain to were rendered together with the 3D brain and tumor ROI

l193 possibly add "deterministic PAS-MRI process in " before "proposed"

l194 delete "in every patient"

l200 the same tracking procedure as what?

l200 adopted, to adopted:

ll201-2 all tracked individual … into a single to the resulting collection of fiber tracks (ie. across tumor (or tumor mirror) and distal seed ROIs) was combined to form a patient-specific connection probability image,

This procedure (as I've understood it) would seem to preclude inclusion in the probability image of pathways through the tumor (or mirror) that don't pass through a distal seed ROI identified with the deterministic method, biasing towards apparent agreement in the results. Might it be better to replicate the distal seed generating process with the probabilistic method, and so have any such missed" projections, one would hen be able to comment on both the relative distributions of tracts and the presence / absence of "missed" tracts. This of course is made problematic by the choice of number of clusters etc which in itself seems subjective. It would be very useful to have a quantitative / statistical comparison of the deterministic results relative to this probability image.
Was the distribution of tracks per region the same As it is, the exert of comparison seems "we looked at them and they were okay" without much validation.


l204 input to the patient's collection of fiber tracts.

l210 contralateral unaffected to unaffected

l219 at tumor to at the tumor

l220 what is under-threshold anisotropy, how would the algorithm have found them - wouldn't FA under some threshold have been a stopping criteria, so did you assess the stoping criteria for all the fibers originating in the distal seed ROIs?

l221-4 delete

l225 three cases to three meningioma cases

l228 In all these cases, meningioma caused a to In all cases of meningioma, we observed a

l229 hemisphere to hemisphere (10 - 35%) - check numbers

l229 architecture to architecture (25 - 98%) - check numbers

l233-5 move this sentence to precede "This indicates…" in l229

l231-3 this material should be moved to the discussion to reflect on the nature / basis for the observations. A reference to support the statement is necessary.

l241 showed to likewise showed

l241 , as well. The to though the

l242 and deviated … intact. -delete.

l242 In patient to Two main connecting bundles were also seen for patient

l243 the inferior to Here however, the inferior

l243 that connected … target ROI - delete

l244 a contralateral - delete

l244 in the lesioned area. to on the lesioned side.

- do any of these tracts have names

l245-6 In all the other cases to In the other low-grade gliomas

l247 pathway but to pathway, but did

l248 tracts that … in the lesioned to tracked fibers seen in the lesioned

l249 comparative - delete

l250 reduced as to reduced on the pathological side

l251 of glioma to of low-grade glioma

l251-2 resulted as being to was

l252 taking into account to indicating a

l254 estimated to involving the tumor

l255 the tracts … target ROI of - delete

l258 In all four cases to Across the four high-grade gliomas

l258 tracts equal to tract fiber counts

probably a bit of care could be given to the naming of what it is tat is reduced - tracts? fiber coutns? bundles? ust one term would be best.

l260 of lesion to , of the percentage lesion

l263 divide the paragraph just before "the connectivity…"

l263 studied by to obtained with

l264 corresponded to to corresponded well with those of our proposed technique with

l264 outputs previously obtained - delete

l264 every patients to every patient

- this part seems to cover that deterministic didn't show extra (would it really be expected?), but was anything missed?


Discussion

Please cite references about the behavior of meningiomas and gliomas in prior literature.

l277 overcame the underestimation - what is the basis for this statement?

l305 - remove page break?

l281 consistently to again, consistent

l282-4 REFERENCE!

l286 you make no mention of role of edema - is it all necessarily infiltration?

l288 an effective to a

l293 The technique we proposed here adopted to In the proposed technique, we adopted - otherwise the technique seems to have a mind of its own…

l294 Nevertheless, to The technique could however, be adapted to use

l294 is an optimal to , which is an optimal

l297 By repeating the comparative … statistical approach. - what is this statistical approach, and ow did it impinge on the results. I would suggest deleting this sentence and unite the paragraphs, usage the sentence:

As part of our evaluation, we compared the results of our technique with those obtained using a probabilistic framework.

l298 This step of the analysis to The latter (this sentence now following the above suggestion)

l300 routine setting to routine setting, though this may change with hardware and algorithmic
improvements.

l300 but in this specific case … results of probabilistic tractography
to In our specific cohort, the results of probabilistic tractography showed our
deterministic results obtained with PAS-MRI to yield substantially equivalent results.

l300-1 further suppored the assessment to The consistency of our results with those of probabilistic tractography
supports the assessment …

l303 suggesting the to and suggests the

l304 critical cases to cases

l304 WM paths and the to WM pathways and

l308 obtained and they to obtained in this preliminary study, and their

l311-5 suggest redoing this paragraph:

The inhomogeneity of tumor locations and sizes creates a difficulty in that some tracts may be intersected by the tumor in mid-tract, whilst others effectively extend in only one direction from the tumor area. To accommodate these two contexts, we incorporated a dual ROI tracking approach between distal ROIs when apparent tract passage was seen in the original tracking from the homologous tumor ROI, and used the distal ROI and the tumor (or tumor homolog) ROI for tracking otherwise. The choice as to which approach to apply however was made on a subjective basis, and a set of objective criteria for determining which to perform in a given case would be desirable. It is unlikely that this aspect of our technique influenced the comparisons performed as the methods were applied bilaterally for comparison at the patient level, and were mixed across the patient groups.

l316-7 All the … Udine. - delete

switch order of the sentences:

In the present work, we didn't take into account pre- or peri-operative functional localization findings. Where such information is available, either from investigations such as fMRI, or intra-operative cortical / subcortical electrodes, the speed of the deterministic approach (or with recalculation of the tract probability distributions) may facilitate the incorporation of such data into the surgical guidance.

l323 data. This was … adopt to data in order to take advantage of the

l324 the packages instead of .. workflow. to the various packages.

l325 a clinical routine to potential use in clinical routine

l328 limitate to limit

l335 Nevertheless, the … introducing a to Comparison with a

l335 framework. to framework showed this not to be a significant problem in our cohort.

l342 best exploit to exploi


References

please check the use of punctuation marks and spacing around the volume(issue) items for all refs.

---

## Author Rebuttal · Round 0.2

**ISTITUTO ITALIANO DI TECNOLOGIA**

Dear Editor,

Thanks for your response and for the reviewer's comments on our manuscript. We did, accordingly, a deep work of revision by modifying the paper in response to the extensive and insightful comments. We have rewritten many sections of the manuscript, changed some figures and we hope that this complies now with the referee's remarks. We respond to the comments point counter point.

All of the changes made with respect to the last version are marked in red.

We hope that now the manuscript could be considered acceptable for publication on PeerJ.

Yours sincerely,

Martina Campanella

(corresponding author)

Fondazione Istituto Italiano di Tecnologia

Sedeegale: Via Morego, 30  16163 Genova    Uffici di Roma: Via Guidubaldo del Monte, 54  00197 Roma
Tel. 010 71781  Fax. 010 720321
C.F. 97329350587 – P.I. 09198791007

[Figure]

**REV 1**

The introduction provides an acceptable general introduction, but fails to clearly identify why an alternative is needed to existing approaches.

*We better clarified this aspect in the Introduction explaining how crucial is to look for solutions different from tractography based on anatomical or functional landmarks in case of white matter architecture disruption.*

The last paragraph of the introduction would benefit greatly from separating the outline of the method from the brief description of how the method is to be evaluated.

*We re-arranged the Introduction by separating the outline of this study from its depiction.*

The rationale of the method needs to be outlined in a suitable manner because the methods section breaks the process into steps that are difficult to follow the logic of the process.

*We organized better the Method session, following the logical process that brings to the definition of the tracts of interest.*

The rationale for and importance of the probabilistic evaluation should also be better stated.

*We explained in a more extensive way the importance of introducing a complementary probabilistic framework to provide a further validation to the deterministic results. The advantages that probabilistic techniques have in some applications are somewhat reduced here, because we ran deterministic tractography from many starting points, which has a similar effect to the stochastic influence in probabilistic tractography. The comparative results we included in the manuscript support our choice of a simpler and quicker deterministic algorithm, more suitable for clinical routine.*

Fondazione Istituto Italiano di Tecnologia

Sedeegale: Via Morego, 30  16163 Genova    Uffici di Roma: Via Guidubaldo del Monte, 54  00197 Roma
Tel. 010 71781  Fax. 010 720321
C.F. 97329350587 – P.I. 09198791007

[Figure]

While informed consent is stated, there is no indication that the study itself had been approved by the responsible ethical committee or research review board.

*We inserted the indication about the ethical approval in the proper section of the Methods.*

In several points, the abstract is vague or imprecise where a bit more clarity would serve.

*We changed the misleading and awkward terms in the Abstract.*

The "Targets" paragraph needs some work to clarify the process.

*We organized better the "Target" paragraph, following the logic process of the Target ROIs generation.*

The method is reasonably described, though the sub-section on tracts of interest was not sufficiently clear for me to understand how the number of centroids generated was decided.

*We clarified this aspect in the Method section, in the Target Generation paragraph.*

The results section needs fairly extensive rewriting. In particular, there are numerous comments that speculate as to the reason behind individual observations (e.g. "probably as a result of the tumor mass", and several other sentences involving "probably"). All of these comments should be eliminated, and such speculation restricted to the discussion (e.g. This could indicate that the underlying axonal structures have remained intact but spatially displaced) with relevant literature cited to support the claims or suggestions. When talking about the various pathways found, it might be useful to indicate where the lesion was and where the observed tract end point centroid(s) were.

*We avoided all the confusing sentences, we re-organized all the section and we moved the comments to the discussion.*

Fondazione Istituto Italiano di Tecnologia

Sedeegale: Via Morego, 30 16163 Genova    Uffici di Roma: Via Guidubaldo del Monte, 54 00197 Roma
Tel. 010 71781 Fax. 010 720321
C.F. 97329350587 – P.I. 09198791007

[Figure]

The discussion also needs considerable revision, firstly to provide a more readily followed line of thought, and more to reflect on its place in context of existing literature. As part of the connection to the literature, it would be good to reflect evidence from elsewhere that disruption is particularly consistent with tumor cell infiltration and not simply edematous changes. There is a large literature on individual diffusion parameters in brain tumors, some relation with how the changes reported in those studies relate to tractography could be attempted.

*We almost completely re-wrote the Discussion section. We commented our findings in contest of the literature and we discuss now more the methodological issues.*

There are a large number of linguistic peculiarities and greater care is needed in providing a consistent and complete message.

*We corrected all the linguistic inaccuracies through the manuscript.*

**REV 2**

The authors use HARDI and PASMRI. Even if there is nothing wrong with that, it may sound as an overkill for patient data. The choice becomes clear later in the manuscript and in the discussion, but it could be clarified and justified here as well. Maybe the authors could briefly mention that despite other techniques being available for resolving crossings using low angular resolution data (e.g. (Peled et al, MRI 2006), (Tournier et al, NeuroImage 2007), (Sotiropoulos et al, JMRI 2008), etc) the particular choice ensures higher sensitivity to fibre crossings.

*We clarified the choice of a multifibre approach in the Introduction section, and better in the Discussion as well, underlining the importance of fully capturing the complex axonal configurations.*

Fondazione Istituto Italiano di Tecnologia

Sedeegale: Via Morego, 30 16163 Genova   Uffici di Roma: Via Guidubaldo del Monte, 54 00197 Roma
Tel. 010 71781 Fax. 010 720321
C.F. 97329350587 – P.I. 09198791007

[Figure]

Probabilistic Tractography: Please provide references (e.g. Behrens et al MRM 2003, Parker et al JMRI 2003).

*We inserted the right references about probabilistic tractography in the Introduction.*

The choice of the imaging protocol is suboptimal. Voxels are anisotropic (suboptimal for tractography, as signal will be lower and therefore uncertainty will be higher along the smaller voxel dimensions). Also, multi-shot acquisitions are more susceptible to subject motion, which is clearly an issue for patients. The authors need to justify these choices and/or at least discuss them as limitations. Finally, the reported resolution is probably interpolated. What is the native resolution of the data? (e.g. 120mm/70 slices does not give 1.5mm that the authors report as slice thickness)

*We corrected the wrong data about the DWI sequence applied (FOV) and we discussed the imaging protocol used in the Limitations of the study section.*

Using five packages to do deterministic tractography is clearly unnecessary. Nothing wrong about that, but you make your life more difficult. Most of the functions you have performed using AFNI/SPM for instance can be done using FSL or SPM alone.

*We discussed this issue in the Limitations of the study section.*

References 9 and 35 are repeated, they are the same.

*We deleted the repeated reference.*

Enhanced / Unenhanced: You mean Gadolinium-enhanced T1? Please clarify.

*We clarified it now in the text.*

[Figure]

Defining tracts of interest. I feel this is an awkward way of defining tracts. Tractography can estimate paths everywhere, so what do the estimated paths in terms of anatomy and their relationship to the tumour area (particularly the ones identified between two targets)? Could you give us some insight behind this choice as opposed to focusing e.g. on specific tracts defined via strictly anatomical criteria?

*We explained better in the manuscript our choice of not using seeds based on anatomical or functional landmarks for tracking (Introduction and in the Discussion sections)*

Figure 5: For the Meningioma case the shown example does not clearly support the main conclusion (i.e. that tracts are displaced by the tumour). Could you please provide a probabilistic tractography result for cases M3 or M4, rather than M2? That would make your point stronger and clearer.

*We provided the connection probability map for case M3, instead of M2.*

## REV 3

The description of the methodology is complex and it should be less redundant. The authors should try to reduce it in length.

*We re-organized the Methods sections, clarifying all the logical thread to the definition of the tracts of interest. Unfortunately we were not able to shorten it significantly.*

The clinical research question is broad and not very well identified.

*We clarified better the main aim of our study, explaining the insight behind our data-driven approach in the Introduction, in the Discussion and in the Limitation sections.*

Fondazione Istituto Italiano di Tecnologia

Sedeegale: Via Morego, 30  16163 Genova    Uffici di Roma: Via Guidubaldo del Monte, 54  00197 Roma
Tel. 010 71781  Fax. 010 720321
C.F. 97329350587 – P.I. 09198791007

[Figure]

The authors made a big effort to use a novel approach with "user-independent" semi-automatic definition of seed and target ROIs, however their only minimal supervised method is going to leave out (or miss) several white matter tracts within and around the mass that may have an important role in brain function.

*The proposed method aims at reconstructing all bundles and trajectories of tumor-involved WM tracts in relation to the contralateral healthy myeloarchitecture without any a-priori anatomical knowledge. We are aware that our approach isn't supervised by an atlas. Nevertheless, we visually inspected every reconstructed trajectory, from the seed tracking to the generation of the tracts of interest. Moreover, we decided to run a complementary probabilistic analysis to confirm the results and modeling their uncertainty. Finally, we used a multi-fibre tracking algorithm in order to overcome the underestimation of the extent of estimated tracts.*

The authors stop short from identifying the name of the major bundles that are infiltrated/damaged by the lesion.

*We now identify the name of the WM fiber tracts, resulting from the tracking analysis.*

The choice to use the PAS functions for multi-fiber reconstruction was a good one, because this method is less sensitive than classic DTI to missing white matter tracts in areas of crossing, branching or fanning. However this method has its own limitations that the authors should acknowledge in the appropriate section of the discussion.

*We now discuss advantages and limitations of the adopted reconstruction algorithm in Discussion and Limitations.*

Fondazione Istituto Italiano di Tecnologia

Sedeegale: Via Morego, 30  16163 Genova    Uffici di Roma: Via Guidubaldo del Monte, 54  00197 Roma
Tel. 010 71781  Fax. 010 720321
C.F. 97329350587 – P.I. 09198791007

[Figure]

The selection of the targets appear to be partially bias. In areas of T2WI signal abnormality diffusion imaging may miss existing fibers. There are asymmetric tracts in the human brain (i.e. arcuate fasciculus). How do the authors controlled for these two issues? They should address this issue in the M&M section.

*We explained better the target generation process, clarifying how we managed the estimation of asymmetric tracts in the Methods section. In addition we commented the possibility of a loss of reconstructed WM bundles in the Discussion and Limitations sections.*

A major limitation of this study was the lack of a reference index such as intraoperative electrostimulation subcortical mapping.

*Unfortunately we did not have extensive subcortical data. In some cases the surgeon stimulated the tissue under the lesion but only when there was the doubt of an excessive proximity to functionally relevant tracts. We engaged this issue in the Limitations section, explaining the choice of a completely data-driven tractography approach.*

LGG are known to infiltrate white matter tracts while HGG tend to dislocate them and eventually destroy them. In the discussion the authors should make a comment about what they found and what it is reported in the literature. The Discussion is weak and very brief. The authors should put their finding in contest of the literature. They should describe in greater detail potential clinical implications of their findings on planning surgery in patients harboring neoplasms infiltrating eloquent areas.

*We re-wrote the Discussion, commenting the obtained alteration patterns in contest of the literature, for the three types of brain tumors evaluated. In addition we better clarified the importance of the suggested method in the clinical environment in the Conclusion section.*

Fondazione Istituto Italiano di Tecnologia

Sedeegale:  Via Morego, 30  16163 Genova    Uffici di Roma:  Via Guidubaldo del Monte, 54  00197 Roma
Tel. 010 71781  Fax. 010 720321
C.F. 97329350587 – P.I. 09198791007

[Figure]

In the Limitation paragraph the authors should acknowledge limitation of tracking with PAS in areas of T2-weighted signal abnormalities. They should also acknowledge possible bias of their methods in ROI target selection. Main WM tracts coursing in the proximity or within the tumor may be missed with the approach used in this study.

*We discussed  these issues in the Limitations sections.*

---

## Round 0.3 · accepted · Accept

This further revision addresses all remaining reviewers' concerns and comments and I am happy to see this work published in PeerJ.